# Electrical Stability Modeling Based on Surface Potential for a-InGaZnO TFTs under Positive-Bias Stress and Light Illumination

**DOI:** 10.3390/mi14040842

**Published:** 2023-04-13

**Authors:** Xiaoming Huang, Wei Cao, Chenyang Huang, Chen Chen, Zheng Shi, Weizong Xu

**Affiliations:** 1College of Integrated Circuit Science and Engineering, Nanjing University of Posts and Telecommunications, Nanjing 210023, China; 1220024107@njupt.edu.cn (W.C.); hcy611@126.com (C.H.); 1021021012@njupt.edu.cn (C.C.); 2School of Communications and Information Engineering, Nanjing University of Posts and Telecommunications, Nanjing 210023, China; shizheng@njupt.edu.cn; 3National Laboratory of Solid State Microstructures, Nanjing University, Nanjing 210093, China; njuphyxwz@nju.edu.cn

**Keywords:** a-IGZO TFTs, sub-gap density of states, surface potential, electrical stability model

## Abstract

In this work, an electrical stability model based on surface potential is presented for amorphous In-Ga-Zn-O (a-IGZO) thin film transistors (TFTs) under positive-gate-bias stress (PBS) and light stress. In this model, the sub-gap density of states (DOSs) are depicted by exponential band tails and Gaussian deep states within the band gap of a-IGZO. Meanwhile, the surface potential solution is developed with the stretched exponential distribution relationship between the created defects and PBS time, and the Boltzmann distribution relationship between the generated traps and incident photon energy, respectively. The proposed model is verified using both the calculation results and experimental data of a-IGZO TFTs with various distribution of DOSs, and a consistent and accurate expression of the evolution of transfer curves is achieved under PBS and light illumination.

## 1. Introduction

Amorphous In-Ga-Zn-O (a-IGZO) thin film transistors (TFTs) have been widely investigated as pixel-switching and driving devices for active-matrix display technology because they offer high-current drive capacity, low off-state power consumption, and low-temperature uniform deposition compared with amorphous silicon-based TFTs [1,2]. However, there are high-density sub-gap defects located in the bandgap of a-IGZO, which cause a change in threshold voltage (Δ*V_th_*) and the degradation of subthreshold swing (Δ*SS*) under electrical and light stress, respectively [3,4]. Correspondingly, the uniformity of pixel-to-pixel brightness are affected in active-matrix display applications [5,6]. As a result, electrical stability modeling is required for the accurate stability prediction of a-IGZO TFTs under external stress conditions.

Up to now, some models for a-IGZO TFTs have been proposed to describe current–voltage characteristics, such as the surface potential ($\varphi_{s}$)-based compact model, charge-based capacitance model and unified dc/capacitance compact model [7,8,9]. Although the band-tail states and localized deep states are considered in these models for a-IGZO TFTs, the localized sub-gap density of states (DOSs) is invaluable in solving the model. Actually, new defects are generated in the device channel or at the a-IGZO/gate dielectric interface under external stress conditions [10,11]. For example, it has been demonstrated that the oxygen interstitial (*O_i_*)-related defects are created in the device channel or the interface region under gate bias stress, originating from weakly bonded oxygen ions in a-IGZO TFTs [12,13]. Furthermore, the occupied deep-level oxygen vacancy (*V_o_*) defects (energy width of ~1.5 eV) near the valence band maximum (VBM) in a-IGZO are ionized into single-ionized *V_o_* (*V_o_^+^*) and double-ionized *V_o_* (*V_o_*^2*+*^)-related defects under the corresponding incident photon energy [4,14]. In this work, to realize the electrical stability modeling for a-IGZO TFTs, the stretched exponential distribution relationship for created *O_i_*-related defects as a function of the PBS time (*t*) and the Boltzmann distribution relation for generated *V_o_*-related defects versus the incident photon energy (*E_ph_*) are adopted to calculate the $\varphi_{s}$ in the model. The accuracy of the proposed model is confirmed by comparison calculation results and the experimental data of a-IGZO TFTs with various fabrication conditions. It is found that the evolution of transfer curves for a-IGZO TFTs under PBS and light stress is in good agreement between the calculated and experimental results.

The back-gate structure of TFTs was applied for the modeling in this work, as shown in Figure 1. SiO_2_ (200 nm) thin film as a gate dielectric layer was grown on a heavily doped n-type Si substrate by plasma-enhanced chemical vapor deposition (PECVD) at a temperature of 300 °C. Next, a 50 nm a-IGZO film was deposited by pulsed laser deposition (PLD) at room temperature, and the oxygen partial pressure (*Po*_2_) was set at 1 Pa and 3 Pa, respectively. The composition of the ceramic target used was In:Ga:Zn = 2:2:1 in an atom ratio. The TFT channel region was then formed by optical photolithography and wet chemical etching. Subsequently, a Ti/Au bilayer was grown by e-beam evaporation for the source/drain metal electrode. The fabricated TFTs had a channel width and length of 200 µm and 40 µm, respectively. Finally, a SiO_2_ (100 nm) thin film used as a passivation layer was grown by PECVD. The samples were annealed in air at T = 300 °C.

## 2. Model Calculation

### 2.1. Surface Potential Model Calculation

For an n-type a-IGZO TFT, the sub-gap trap distribution comprises both the Gaussian deep states and exponential tails near the conduction band (*E_C_*) edge [15,16]. Based on the gradual channel approximation, the one-dimensional Poisson’s equation is given by
(1)$$\frac{d^{2}\varphi }{dx^{2}}=\frac{q}{\varepsilon_{\mathrm{igzo}}}\left(n_{\mathrm{free}}+n_{\mathrm{tail}}+n_{g}\right)$$
where the *x*-direction is defined as perpendicular to the channel direction, as shown in Figure 1. φ is the electrostatic potential, q is the electric charge, and $\varepsilon_{\mathrm{igzo}}$ is the permittivity of a-IGZO. The free-electron concentration (*n_free_*) in the channel, the electron concentration of tail states (*n_tail_*), and the electron concentration of deep states (*n_g_*) can be expressed as follows:(2)$$n_{\mathrm{free}}=n_{i}\exp\left[\frac{E_{f}+q\left(\varphi -V_{\mathrm{ch}}\right)}{q\phi_{f}}\right]$$
(3)$$n_{\mathrm{tail}}=N_{T}\exp\left[\frac{E_{f}-E_{c}+q\left(\varphi -V_{\mathrm{ch}}\right)}{w_{\mathrm{tl}}}\right],$$
(4)$$n_{g}=\sum_{k=1}^{n}\frac{dN_{k}}{1+\frac{1}{g}e^{\frac{E_{k}-E_{f}-q\varphi }{q\phi_{f}}}},$$
where *n_i_* is the intrinsic electron concentration, $\phi_{f}$ is the thermal voltage, *V_ch_* is the channel potential, and *E_f_* is the Fermi-level energy; *N_T_ = g_tail_*(π*kT/sin*(π*kT/w_tl_*)) when *w_tl_* > *kT* [17], herein, *g_tail_* is the tail states density at *E_C_*, and *w_tl_* is the conduction band tail slope. In this work, to solve the integration of Equation (4), the rectangle rule is applied to subdivide an integral interval into *n* equal rectangle levels [17]. In Equation (4), $N_{k}=N_{G}\exp\left[-{\left(E_{k}-E_{0}\right)}^{2}/{\left(q\phi_{g}\right)}^{2}\right]$, where *N_G_* is the peak value of the deep states, *E_k_* is the center energy value of each discrete rectangle distribution, *E*_0_ is the energy value within the band gap corresponding to the peak of the deep states, *q*$\phi_{g}$ is the characteristic decay energy of the deep states, *d* is the distance between every two discrete levels, and *g* is defined as the degenerescence constant. By combining Equation (1) with the derivative relation, $2\left(d\varphi /\mathrm{dx}\right)\left(d^{2}\varphi /dx^{2}\right)=\left(d/\mathrm{dx}\right){\left(d\varphi /\mathrm{dx}\right)}^{2}$, the magnitude of the electric field (*E*(φ) = *d*φ/*dx*) can be written as
(5)$$\frac{d\varphi }{\mathrm{dx}}=\sqrt{\frac{2q}{\varepsilon_{\mathrm{igzo}}}\int_{0}^{\varphi }\left(n_{\mathrm{free}}+n_{\mathrm{tail}}+n_{g}\right)d\varphi }.$$

By using Gauss’s law at the TFT interface region, the $\varphi_{s}$ can be expressed by
(6)$$V_{\mathrm{GS}}-V_{\mathrm{FB}}-\varphi_{s}=\frac{\varepsilon_{\mathrm{igzo}}E\left(\varphi_{s}\right)}{C_{\mathrm{OX}}},$$
where *V_GS_* is the gate-source voltage, *V_FB_* is the flat-band voltage, and *C_OX_* is the oxide capacitance per unit area. Then, substituting Equation (5) into Equation (6), the relationship between the *V_GS_* and $\varphi_{s}$ can be written as
(7)$$V_{\mathrm{GS}}-V_{\mathrm{FB}}-\varphi_{s}=\frac{\sqrt{2q\varepsilon_{\mathrm{igzo}}}{\left\{N_{f}\left[\exp\left(\varphi_{s}/\phi_{f}\right)-1\right]+N_{\mathrm{tl}}\left[\exp\left({q\varphi }_{s}/w_{\mathrm{tl}}\right)-1\right]+N_{g}\right\}}^{\frac{1}{2}}}{C_{\mathrm{OX}}},$$
in the above Equation (7), $N_{f}=n_{\mathrm{free}}\phi_{f}\exp\left(-\varphi /\phi_{f}\right)$, $N_{\mathrm{tl}}=n_{\mathrm{tail}}\phi_{\mathrm{tl}}\exp\left(-q\varphi /w_{\mathrm{tl}}\right)$, and $N_{g}=\phi_{f}\;\sum_{k=1}^{n}N_{k}\ln\left[C_{k}+\exp\left(\varphi_{s}/\phi_{f}\right)\right]$. Herein, $C_{k}=\frac{1}{g}\times \;\exp\left[\left(E_{k}-E_{f}\right)/q\phi_{f}\right]$.

Using charge-sheet approximation and then taking both the diffusion and drift current into account, the drain current can be expressed as
(8)$$I_{\mathrm{DS}}\left(y\right)=\mu_{E}W\left(\phi_{f}\frac{dQ_{i}}{\mathrm{dy}}-Q_{i}\frac{d\varphi_{s}}{\mathrm{dy}}\right),$$
where $\mu_{E}$ is the electron mobility, $\mu_{E}=\mu_{i}{\left(V_{\mathrm{GS}}-V_{\mathrm{FB}}\right)}^{p}$, *μ_i_* and *p* are the fitting parameters, *W* is the channel width, *y*-direction is parallel to the device interface, and *Q_i_* is the induced charge density per unit area: *Q_i_ =* −*C_OX_*(*V_GS_* − *V_FB_* − $\varphi_{s}$) − *Q_tail_* − *Q_g_*. Herein, the charge density of the tail state (*Q_tail_*) and the charge density of the deep states (*Q_g_*) are given by
(9)$$Q_{\mathrm{tail}}=\int_{0}^{t_{\mathrm{igzo}}}-qn_{\mathrm{tail}}\mathrm{dx}=-qn_{\mathrm{igzo}}t_{\mathrm{igzo}},$$
(10)$$Q_{g}=\int_{0}^{t_{\mathrm{igzo}}}-qn_{g}\mathrm{dx}=-qn_{g}t_{\mathrm{igzo}},$$
and consequently, Equation (8) can be obtained by integrating this along the *y*-direction. The result can be expressed as
(11)$$I_{\mathrm{DS}}=\mu_{E}\frac{W}{L}\left\{\phi_{f}\left[Q_{i}\left(\varphi_{\mathrm{sd}}\right)-Q_{i}\left(\varphi_{\mathrm{ss}}\right)\right]-\int_{\varphi_{\mathrm{ss}}}^{\varphi_{\mathrm{sd}}}Q_{i}\left(\varphi_{s}\right)d\varphi_{s}\right\}$$
where *L* is the channel length, and $\varphi_{\mathrm{sd}}$ and $\varphi_{\mathrm{ss}}$ are the value of $\varphi_{s}$ when *V_ch_* = *V_DS_* and *V_ch_* = 0 V, respectively. In Equation (11), the integral result of *Q_i_*($\varphi_{s}$) can be written as
(12)$$G(\varphi_{s})=qt_{\mathrm{igzo}}[\phi_{\mathrm{tail}}n_{\mathrm{tail}}+\phi_{f}\sum_{k=1}^{\;n}N_{k}\ln\left(C_{k}+e^{\frac{\varphi_{s}}{\phi_{f}}}\right)]$$

Finally, substituting Equation (12) with Equation (11), the drain current (*I_DS_*) equation can be obtained as
(13)$$I_{\mathrm{DS}}=\mu_{E}\frac{W}{L}\left\{\phi_{f}\left[Q_{i}\left(\varphi_{\mathrm{sd}}\right)-Q_{i}\left(\varphi_{\mathrm{ss}}\right)\right]-\left[G\left(\varphi_{\mathrm{sd}}\right)-G\left(\varphi_{\mathrm{ss}}\right)\right]\right\}.$$

### 2.2. Model Verification

To confirm the accuracy of the model, the $\varphi_{s}$ solution is compared with numerical results with different *V_ch_* values and defect distributions.

As shown in Figure 2a,b, it was found that the surface potential characteristics are in line with numerical results. The parameters for the model calculation are summarized in Table 1. Meanwhile, to further evaluate the model validity, a comparison of the model results with the measured experimental data from a-IGZO TFTs fabricated with *Po*_2_ values of 1 Pa and 3 Pa was carried out. It has been demonstrated that DOSs mainly originate from *V_o_*-related traps in a-IGZO, and the distribution of DOSs is affected by the *Po*_2_ during channel layer deposition [18,19]. Correspondingly, the quantity of *V_o_* within the a-IGZO films grown with *Po*_2_ values of 1 Pa and 3 Pa was analyzed by X-ray photoelectron spectroscopy (XPS). As shown in Figure 3a,b, the binding energy peaks at 530.1 eV, 530.9 eV, and 532 eV are associated with the *O*^2*−*^ ions combined with metal atoms (*M-O*), *V_o_*, and *OH^−^*, respectively [20,21]. The relative amount of *V_o_* in the a-IGZO films could be described by the area ratio of the *V_o_* peak to the total O 1s peak, which are 37% and 32% for a-IGZO films grown with *Po*_2_ values of 1 Pa and 3 Pa, respectively. Therefore, the output and transfer characteristics of devices fabricated with *Po*_2_ values of 1 Pa and 3 Pa are used to compare with those of the model. As shown in Figure 4a,b, the model results are in accordance with the measured I–V curves for a-IGZO TFTs. The obtained model parameters are presented in Table 1. Thus, these results confirm that the model can accurately predict the I–V characteristics for a-IGZO TFTs with various distributions of DOSs. In addition, it has been reported that new trap states are generated for a-IGZO TFTs under PBS and light stress, which cause the diminishment in electrical performance [10,22,23]. Therefore, in the following, the newly created defect DOS model is used to describe the electrical stability of a-IGZO TFTs under external stress conditions.

## 3. Results and Discussions

### 3.1. The Model of a-IGZO TFTs under PBS

It has been demonstrated that the Δ*V_th_* values for a-IGZO TFTs under PBS are mainly ascribed to the creation of oxygen-related traps [10,24]. Under PBS, the oxygen interstitials (*O_i_*) are generated from the weakly bonded oxygen ions in a-IGZO, which are in an octahedral configuration [*O_i_*(*oct*)] and are electrically active. The created *O_i_*(*oct*)-related defects located above the mid-gap are occupied by trapping electrons and become the negatively charged defect [*O_i_*^2*−*^(*oct*)] when the *E_f_* increases under PBS, which causes a positive drift in the transfer curve of a-IGZO TFTs [24,25]. Meanwhile, the generated *O_i_*^2*−*^(*oct*)-related defects are transformed into deep-level negative-U states due to the structural relaxation effect. Because the generated *O_i_*(*oct*)-related defects are distributed above the mid-gap, the peak value of the Gaussian-distributed deep defects are adjusted to describe the generation of new defects of a-IGZO TFTs during the PBS process. In previous reports, the Δ*V_th_* in a-IGZO TFTs under PBS depended on the characteristic trapping time of carriers (*τ*) and the dispersion parameter of the barrier energy height (*β*), which is in accordance with the stretched exponential distribution relationship [26,27]. Therefore, in this work, the amount of created *O_i_*-related traps (Δ*N_G_*) with the increase of PBS time (*t*) can be expressed as follow:(14)$$\Delta N_{G}=\alpha V_{0}\{1-\exp\left[-{\left(\frac{t}{\tau }\right)}^{\beta }\right]\}$$
where *α* is the fitting parameter relating to the DOSs, and *V*_0_ is *V_bias_*-*V*_*th*0_ (*V_bias_* and *V*_*th*0_ are the PBS voltage and the initial *V_th_* of the device, respectively).

Next, substituting Equation (14) into Equation (3), the modified DOSs model can be written as
(15)$$N_{k}=\left(N_{G}+\Delta N_{G}\right)\exp\left[-{\left(\frac{E_{k}-E_{0}}{q\phi_{g}}\right)}^{2}\right].$$

Finally, the *I_DS_* of a-IGZO TFTs under PBS could be calculated by using the proposed DOSs model.

To validate the proposed scheme, the model is compared with the experimental results from the evolution of transfer curves versus PBS time for devices fabricated with *Po*_2_ values of 1 Pa and 3 Pa, respectively. The PBS stability of the a-IGZO TFTs is measured at a *V_GS_* of 20 V for the PBS time of 5000 s. Based on the proposed DOS model calculation, it is clear that the peak value of the Gaussian-distributed oxygen-related defects (*N_G_*) is increased as the PBS time increases for the a-IGZO TFTs, as shown in Figure 5a,b. Meanwhile, the Δ*N_G_* after 5000 s of PBS is 1.05 × 10^18^ cm^−3^eV^−1^ and 6.8 × 10^17^ cm^−3^eV^−1^ for a-IGZO TFTs with *Po*_2_ values of 1 Pa and 3 Pa, respectively. In addition, as shown in Figure 6a,b, it is found that the evolution of transfer curves for the devices calculated using the defect DOS model agrees well with the experimental results, and the specific values of the relevant parameters are listed in Table 2. Correspondingly, the Δ*V_th_* values are 4.5 V and 2.5 V for a-IGZO TFTs fabricated with *Po*_2_ values of 1 Pa and 3 Pa, suggesting that the oxygen-related defects are suppressed in the channel by the increasing in *Po*_2_. This result can also be supported by XPS analysis. Thus, the presented results confirm that the model could accurately predict the PBS stability of a-IGZO TFTs.

### 3.2. The Model of a-IGZO TFTs under Light Illumination

It has been demonstrated that there are high-density entirely occupied *V_o_* values existing in the VBM with the energy distribution of ~1.5 eV, which are ionized to *V_o_^+^* and *V_o_*^2*+*^ under short-wavelength light. During photo-excited ionization processes, the new unoccupied defect states are generated at the bottom of *E_C_* and mid-gap due to the outward lattice relaxation effect [28,29]. Meanwhile, the values of the activation energy (*E_a_*) for the photo-induced transition from *V_o_* to *V_o_^+^* and *V_o_*^2*+*^ are ~2.0 eV and ~2.3 eV, respectively. Figure 7a shows the process of generating *V_o_*-related traps in a-IGZO under monochromatic light. Since the generated *V_o_*-related defects are mainly distributed near the bottom of the *E_C_* [29,30], the conduction band tail slope is adjusted to describe the creation of *V_o_*-related defects for a-IGZO TFTs under a light illumination process. It has been reported that the exponential band tails can be expressed by a generalization of Boltzmann relations in semiconductor materials [31,32]. Thus, in this work, the Boltzmann distribution relation is applied to describe the generated *V_o_*-related defects at the bottom of the *E_C_* under light conditions. The amount of change in the conduction band tail slope (Δ*w_tl_*) versus the incident photon energy (*E_ph_*) can be written as
(16)$$\Delta w_{\mathrm{tl}}=\sigma \{1-{\left[1+\exp(\frac{E_{\mathrm{ph}}-E_{a}}{\gamma })\right]}^{-1}\}$$
where *σ* is the fitting parameter, and *γ* is the fitting parameter in proportion to the oxygen partial pressure. Substituting Equation (16) with Equation (4), the density of the tail state (*N_TL_*) model of a-IGZO TFTs under light stress can be given by
(17)$$N_{\mathrm{TL}}=g_{\mathrm{tail}}\exp\left(\frac{E-E_{c}}{w_{\mathrm{tl}}+\Delta w_{\mathrm{tl}}}\right).$$

As a result, the *I_DS_* of a-IGZO TFTs under light conditions could be calculated by using the proposed DOS model. To check the proposed scheme, a comparison between the model and the measured transfer curves of a-IGZO TFTs under monochromatic light at various wavelengths for 120 s was carried out. According to the proposed DOS model calculation, it is clear that the Δ*w_tl_* is increased as the wavelength of incident light is reduced from 700 nm to 500 nm for the a-IGZO TFTs, as shown in Figure 7b,c. Correspondingly, the Δ*w_tl_* values are 0.085 eV and 0.043 eV for a-IGZO TFTs with *Po*_2_ values of 1 Pa and 3 Pa after *λ* = 500 nm, as shown in Figure 7d. Meanwhile, the evolution of transfer curves for a-IGZO TFTs calculated using the proposed DOS model is in good accordance with the experimental results, as shown in Figure 8a,b. The specific values of the relevant parameters are listed in Table 3. Under light illumination, the transfer curves of TFTs drift toward the negative direction as the incident wavelength light reduces from 700 nm to 500 nm, which are induced by the photo-excited electrons from the occupied interface defects and occupied deep *V_o_* states. In addition, it has been demonstrated that the Δ*SS* is associated with the amount of created defects (Δ*N_t_*) within the device’s active region [33,34]. It is found that the Δ*SS* values for devices fabricated with *Po*_2_ values of 1 Pa and 3 Pa are 0.9 V/dec and 0.5 V/dec after *λ* = 500 nm, suggesting that deep-level *V_o_* can be suppressed by increasing *Po*_2_. Therefore, these results demonstrate that the model could accurately predict the light stability of a-IGZO TFTs.

## 4. Conclusions

In this work, an electrical stability model based on surface potential for a-IGZO TFTs considering both exponential band tails and Gaussian deep states has been proposed. In this model, a reasonable method is presented for solving the surface potential by using the stretched exponential and Boltzmann distribution relation to describe the generated defects as a function of external stress conditions, the feasibility of which has been demonstrated by achieving good agreement between calculation results and experimental data of a-IGZO TFTs under PBS and light conditions. Thus, the proposed model is accurate and useful for predicting the electrical stability of a-IGZO TFTs.

## Figures and Tables

**Figure 1 micromachines-14-00842-f001:**
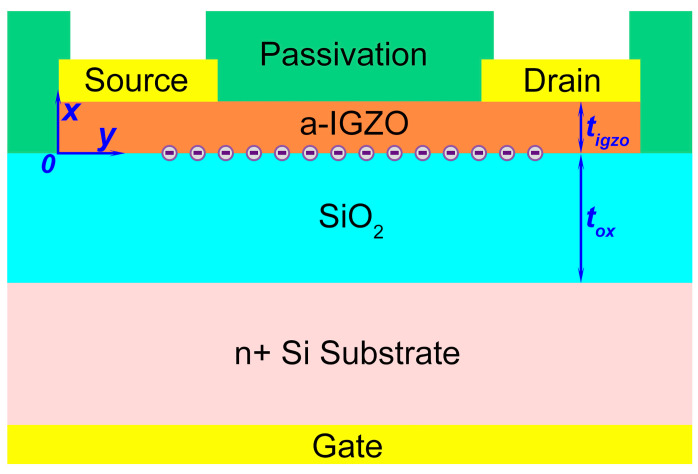
Schematic diagram of a-IGZO TFTs used in modeling.

**Figure 2 micromachines-14-00842-f002:**
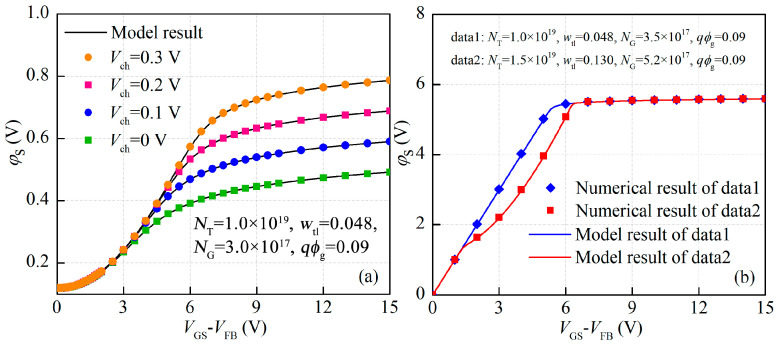
Comparison of the surface potential solution with the numerical result for (**a**) different values of *V_ch_* and (**b**) various DOS distributions.

**Figure 3 micromachines-14-00842-f003:**
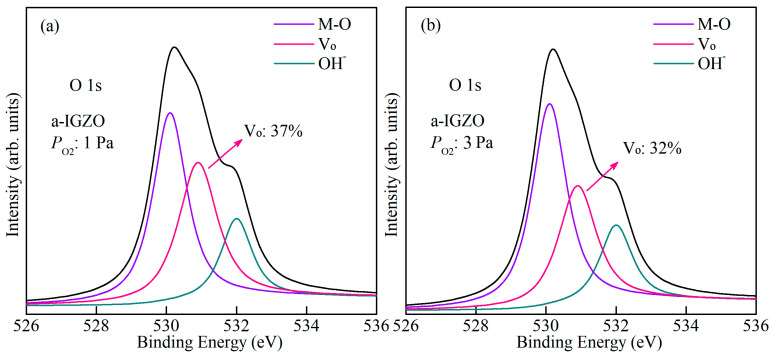
O 1s XPS spectra of a-IGZO thin films fabricated with *Po*_2_ values of (**a**) 1 Pa and (**b**) 3 Pa.

**Figure 4 micromachines-14-00842-f004:**
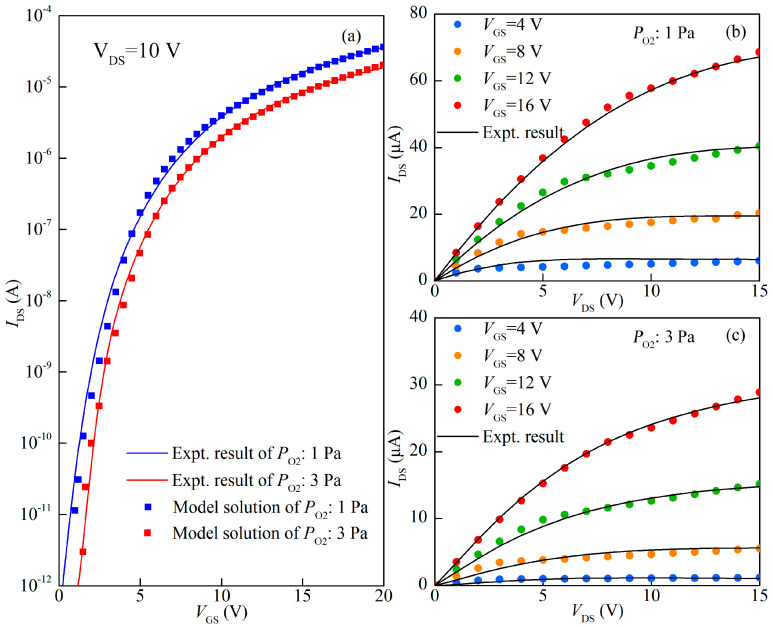
Comparison of (**a**) transfer characteristics and (**b**,**c**) output characteristics for a-IGZO TFTs fabricated with *Po*_2_ values of 1 Pa and 3 Pa between the model results and experimental data.

**Figure 5 micromachines-14-00842-f005:**
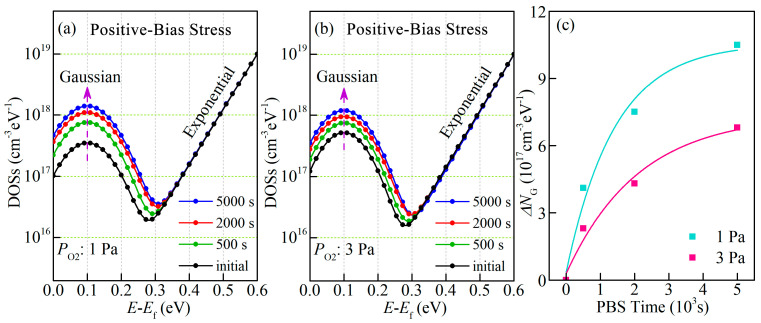
Calculated DOSs as a function of PBS time for a-IGZO TFTs fabricated with different *Po*_2_ values: (**a**) 1 Pa and (**b**) 3 Pa; (**c**) the shift of *N_G_* for a-IGZO TFTs fabricated with *Po*_2_ values of 1 Pa and 3 Pa.

**Figure 6 micromachines-14-00842-f006:**
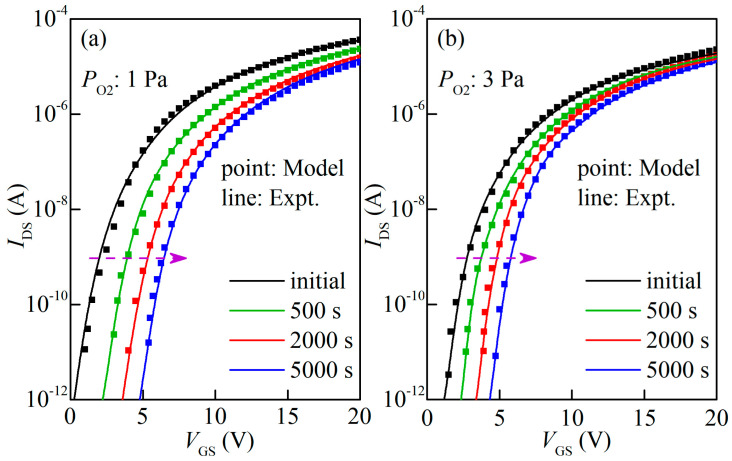
Measured evolution of the transfer curves as a function of PBS time for a-IGZO TFTs fabricated with different *Po*_2_ values compared with model results: (**a**) 1 Pa and (**b**) 3 Pa.

**Figure 7 micromachines-14-00842-f007:**
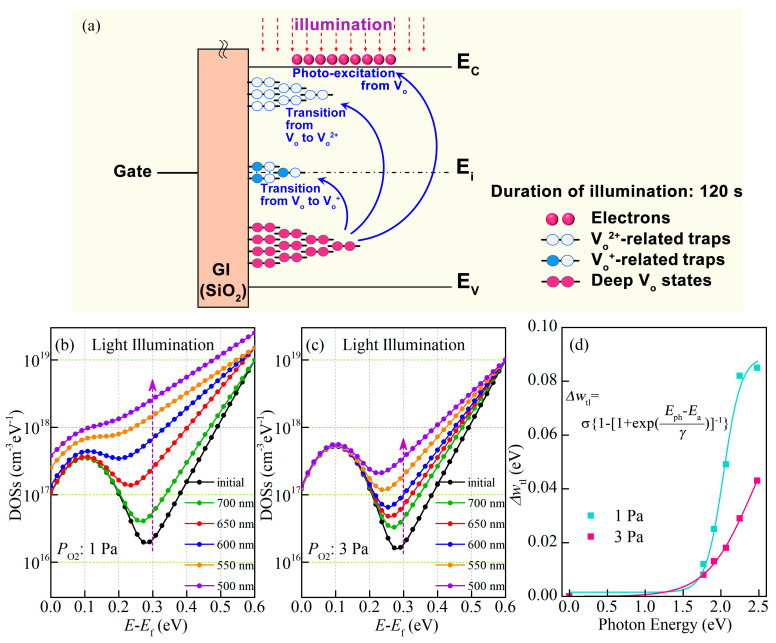
(**a**) Schematic diagram of the *V_o_*-related defect generation process for a-IGZO TFTs under short-wavelength light. Calculated DOSs as a function of monochromatic light illumination for a-IGZO TFTs fabricated with different *Po*_2_ values: (**b**) 1 Pa and (**c**) 3 Pa; (**d**) the shift of *w_tl_* for a-IGZO TFTs fabricated with *Po*_2_ values of 1 Pa and 3 Pa.

**Figure 8 micromachines-14-00842-f008:**
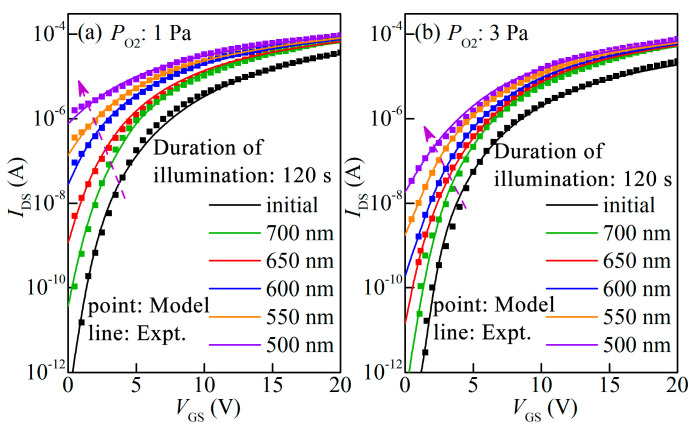
Measured evolution of transfer curves as a function of various conditions of monochromatic light illumination for a-IGZO TFTs fabricated with different *Po*_2_ values compared with model results: (**a**) 1 Pa and (**b**) 3 Pa.

**Table 1 micromachines-14-00842-t001:** Key parameters used in the model.

Parameters	Value	Unit	Parameters	Value	Unit
*q*	1.6 × 10^−19^	C	*d*	0.05	eV
*ε_igzo_*	11.5	-	*E* _0_	2.7	eV
*C_OX_*	1.73 × 10^−4^	F/m^2^	*μ_i_*	5	cm^2^s^−1^/V^1+p^
*V_FB_*	0	V	*p*	0.7	-
*V_DS_*	10	V	*W*	200	µm
*ϕ_f_*	0.026	V	*L*	40	µm
*E_f_*	2.6	eV	*t_igzo_*	50	nm

**Table 2 micromachines-14-00842-t002:** Parameters of the DOSs under PBS.

Parameter	*Po*_2_ [Pa]	Initial	PBS Time [s]		
			500	2000	5000
*N_T_* [cm^−3^eV^−1^]	1	1 × 10^19^	1 × 10^19^	1 × 10^19^	1 × 10^19^
	3	1 × 10^19^	1 × 10^19^	1 × 10^19^	1 × 10^19^
*w_tl_* [eV]	1	0.048	0.048	0.048	0.048
	3	0.047	0.047	0.047	0.045
*N_G_* [cm^−3^eV^−1^]	1	3.5 × 10^17^	7.6 × 10^17^	1.1 × 10^18^	1.4 × 10^18^
	3	5.2 × 10^17^	7.5 × 10^17^	9.5 × 10^17^	1.2 × 10^18^
*qϕ_g_* [eV]	1	0.09	0.09	0.095	0.095
	3	0.083	0.085	0.09	0.09
*β*	1	0.85			
	3				
*τ*[s]	1	1544.58			
	3	1902.39			
*α*	1	6.43 × 10^16^			
	3	5.49 × 10^16^			

**Table 3 micromachines-14-00842-t003:** Parameters of the DOSs under illumination.

Parameter	*Po*_2_ [Pa]	Initial	Wavelength of Light [nm]				
			750	650	600	550	500
*N_T_* [cm^−3^eV^−1^]	1	1 × 10^19^	1 × 10^19^	1.5 × 10^19^	1.5 × 10^19^	1.5 × 10^19^	2.5 × 10^19^
	3	1 × 10^19^	1 × 10^19^	1 × 10^19^	1 × 10^19^	1 × 10^19^	1 × 10^19^
*w_tl_* [eV]	1	0.048	0.060	0.073	0.097	0.130	0.133
	3	0.047	0.055	0.060	0.065	0.076	0.090
*N_G_* [cm^−3^eV^−1^]	1	3.5 × 10^17^	3.5 × 10^17^	3.5 × 10^17^	3.5 × 10^17^	3.5 × 10^17^	3.5 × 10^17^
	3	5.2 × 10^17^	5.2 × 10^17^	5.2 × 10^17^	5.2 × 10^17^	5.2 × 10^17^	5.2 × 10^17^
*qϕ_g_* [eV]	1	0.09	0.09	0.09	0.09	0.09	0.09
	3	0.083	0.083	0.083	0.083	0.083	0.083
*E_a_* [eV]	1	2.3					
	3						
*γ*	1	0.1					
	3	0.3					
*σ*	1	0.089					
	3	0.076					

## Data Availability

The data presented in this study are available on request from the corresponding author.

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
