# Peer review of "Electrical Stability Modeling Based on Surface Potential for a-InGaZnO TFTs under Positive-Bias Stress and Light Illumination"

_micromachines, 2023, doi:10.3390/mi14040842_

Round 1

Author Response

Please see the atttachment.

Reviewer 2 Report

In this research, the authors reported the study of an electrical stability model based on the surface potential for a-IGZO TFTs, including both exponential band tails and Gaussian deep states, to describe the generated defects as a function of external stress conditions. Consequently, the following major comments/questions must be addressed in the revised paper to publish this manuscript in Micromachines. 

1.     When addressing the exploitation of these semiconductors in amorphous In-Ga-Zn-O transistors, please add more research papers associated with the small exploitations for electronics recently published. (Kim et al. Solid-State Electronics 201 (2023) 108605 and Kim et al. Advanced Materials Interface 9, 10 (2022) 2200032)

2.     Please identify the composition/stoichiometry of the IGZO films, as it is quite relevant as far as the device performances and characteristics are concerned.

3.     Please comment on the stability of the films. Also, in the data shown, how reproducible are they and do they correspond to the average values recorded (from different devices batch to batch) or just to the measurements of a single device? It is recommended to provide error bars and standard deviation values from reproducible measurements to improve the reliability of data and simulation fitting results.

4.     IGZO devices typically exhibit typical hysteresis from the forward/reverse bias. Can you consider not only a single forward bias, but also the case of applying a reverse bias?

5.     It would be great if you could briefly chart the clear differences between your study and other representative stability models in existence.

6.     In Figure 7(a), please enlarge the schematic diagrams with clear letter font sizes. In Figure 7(d), the equation should be described in letters in printed form, not in italic form. 

Round 2

Reviewer 2 Report

It would be nice to see a literature survey table to emphasize the differences with other simulation models. I believe that would encourage more relevant researchers to search for this research article.